# Recent Insight into Lipid Binding and Lipid Modulation of Pentameric Ligand-Gated Ion Channels

**DOI:** 10.3390/biom12060814

**Published:** 2022-06-10

**Authors:** Anna Ananchenko, Toka O. K. Hussein, Deepansh Mody, Mackenzie J. Thompson, John E. Baenziger

**Affiliations:** Department of Biochemistry, Microbiology, and Immunology, University of Ottawa, 451 Smyth Road, Ottawa, ON K1H 8M5, Canada; aanan080@uottawa.ca (A.A.); thuss090@uottawa.ca (T.O.K.H.); dmody103@uottawa.ca (D.M.); mthom214@uottawa.ca (M.J.T.)

**Keywords:** pentameric ligand-gated ion channels, lipid binding sites, lipid–protein interactions, annular, non-annular, allosteric modulation

## Abstract

Pentameric ligand-gated ion channels (pLGICs) play a leading role in synaptic communication, are implicated in a variety of neurological processes, and are important targets for the treatment of neurological and neuromuscular disorders. Endogenous lipids and lipophilic compounds are potent modulators of pLGIC function and may help shape synaptic communication. Increasing structural and biophysical data reveal sites for lipid binding to pLGICs. Here, we update our evolving understanding of pLGIC–lipid interactions highlighting newly identified modes of lipid binding along with the mechanistic understanding derived from the new structural data.

## 1. Introduction

Pentameric ligand-gated ion channels (pLGICs) mediate or modulate fast synaptic communication in the central and peripheral nervous systems making them vital for neurological processes ranging from memory and learning to nicotine addiction [1,2,3,4]. pLGICs respond to the binding of neurotransmitters by transiently opening either cation- or anion-selective ion channels across the post-synaptic membrane, with prolonged exposure favoring a non-conductive desensitized state(s). The relative stabilities of the resting, open and desensitized states, as well as the rates of inter-conversation between them, shape the magnitude and temporal nature of the agonist-induced response to establish effective inter-neuronal or neuromuscular communication. pLGICs are also targeted by a variety of exogenous molecules that allosterically modulate the agonist-induced response in a manner that alters synaptic communication [5,6].

Lipids are potent activators and/or modulators of ion channels including inward rectifying potassium channels, voltage-gated potassium channels, transient receptor potential channels, mechanosensitive ion channels and pLGICs [7]. The functional sensitivity of pLGICs to membrane lipids was first shown in the 1970s through studies that sought to identify the structural features in the muscle-type *Torpedo* nicotinic acetylcholine receptor (nAChR) that are responsible for both agonist binding and channel gating. These studies showed that to retain both binding and gating, cholate-solubilized receptors must be purified in the presence of lipids and then placed in a bilayer with a particular lipid composition [8,9]. Since then, the effects of lipids on *Torpedo* nAChR function have been characterized extensively [10,11,12]. More recently, studies of pLGIC–lipid interactions have extended to other members of the super-family, including the prokaryotic *Gleobacter violaceus* ligand-gated ion channel, GLIC, and *Erwinia chrysanthemi* (now *Dickeya dadantii*) ligand gated ion channel, ELIC.

Over the past 15 years, increasing numbers of structures have shed light on the modes of lipid binding to pLGICs, thus providing a structural context for interpreting functional data on pLGIC–lipid interactions [13]. Here, we review our evolving understanding of the mechanisms by which lipids alter pLGIC function highlighting new insight along with gaps in our knowledge.

### 1.1. pLGIC Structure

Both eukaryotic and prokaryotic pLGICs exhibit a common architecture consisting of five subunits arranged either symmetrically (homomeric) or pseudo-symmetrically (heteromeric) around a central ion channel pore (Figure 1). In humans, there are four main families of pLGICs that conduct either cations or anions leading to either excitatory or inhibitory post-synaptic responses, respectively. The excitatory cation-selective pLGICs respond to the neurotransmitters, acetylcholine (nicotinic acetylcholine receptor, nAChR) and serotonin (serotonin receptor, 5-HT_3_R), while the inhibitory anion-selective pLGICs respond to γ-aminobutyric acid (GABA receptor, GABA_A_R) and glycine (glycine receptor, GlyR). Humans also uniquely express a cation-selective zinc-activated channel, ZAC [14]. Each of the four main families includes a variety of functional hetero- and homo-pentamers that form from different combinations of the sixteen distinct nAChR subunits (α1–α7, α9–α10, β1–β4, δ, γ, and ε), the five distinct 5-HT_3_AR subunits (A–E), the nineteen distinct GABA_A_R subunits (α1–α6, β1–β3,γ1–γ3, ε, δ, π, θ, and ρ1–ρ3) or the five distinct GlyR subunits (α1–α4, and β). Each combination leads to a pLGIC with a unique electrophysiological and pharmacological fingerprint. Receptors with different subunit are also targeted to specific cell types and/or regions of the brain [15].

The core of each pLGIC structure consists of an N-terminal extracellular domain (ECD), which typically defines agonist binding, and a transmembrane domain (TMD), which contains both the ion-selective pore and the channel gate. In each subunit, the ECD contains ten β-strands (β1–β10) that form two β-sheets folded together into a β-sandwich. The TMD from each subunit is formed from four transmembrane 𝛼-helices (M1 to M4), with M2 lining the channel pore, M1 and M3 shielding M2 from the surrounding lipids, and M4 interacting extensively with the lipid bilayer. Human pLGICs exhibit an intracellular domain (ICD), located between M3 and M4, that interacts with the cytoskeleton. In cation-selective pLGICs, the ICD starts with a short MX 𝛼-helix oriented parallel to the bilayer surface that participates in lipid binding, followed by a mainly disordered region and then a long amphipathic 𝑎-helix, termed MA, which is contiguous with M4 in most cation-selective pLGICs [16]. In anion-selective pLGICs, electron density has not yet been observed for the ICD, making it uncertain whether the MX and MA 𝛼-helices are conserved in these pLGICs [17,18]. Prokaryotic pLGICs lack the ICD, but in some cases contain extra N-terminal domains located just prior to the ligand-binding ECD. Many of these N-terminal domains exhibit sequence similarities to periplasmic binding proteins, possibly allowing these pLGICs to participate in chemotaxis and/or quorum sensing [19].

Agonist sites are typically formed from a series of loops in the ECD extending from the interfaces between the principal and complementary subunits (Figure 1). Agonist binding leads to a compression of these loops around the bound agonist, which drives a conserved rocking motion of the adjacent β-sandwich. The agonist-induced motions of the β-sandwich ultimately translate to the TMD through the covalent link between β10 and M1 and through non-covalent interactions between the β1–β2, β6–β7, and β9–β10 loops and the M2–M3 linker to open the channel gate [20,21,22]. In the resting state, conserved hydrophobic residues in the extracellular half of M2 create an unsolvated energetic barrier that prevents ion flux into the cell [23]. Concerted movements at the ECD–TMD interface lead to a tilting and twisting of the M2 pore lining α-helices away from the central pore axis, thus widening the hydrophobic barrier to allow the diffusion of hydrated ions down their electrochemical gradient [24,25,26,27,28].

Prolonged exposure of pLGICs to agonist leads to the formation of a desensitized state(s) that binds agonist with high affinity, but does not open in response to agonist binding [29]. In anion-selective pLGICs, desensitization arises from a constriction of a gate located near the intracellular end of the transmembrane pore [30]. Loops at the interface between the ECD and TMD and residues near the desensitization gate both govern the rates of pLGIC desensitization [30,31]. Of particular relevance to this review, lipid binding adjacent to M4 influences the rates of desensitization in the prokaryote, ELIC (see below) [32].

### 1.2. Nicotinic Acetylcholine Receptors

#### 1.2.1. Functional Sensitivity of the nAChR to Lipids

Cell-based assays and mutagenesis indirectly suggest that numerous members of the nAChR family exhibit a functional sensitivity to lipids [33,34,35,36,37,38,39,40]. In addition, functional measurements using the *Torpedo* nAChR reconstituted into liposomes with defined lipid compositions show definitively that a broad range of lipids influence the agonist-induced response, and do so through complex mechanisms. For example, ternary lipid mixtures containing phosphatidylcholine (PC) and both cholesterol and anionic lipids support a robust agonist-induced response, while PC membranes lacking both lipids lock the nAChR in a non-responsive *uncoupled* conformation that binds agonist but does not normally undergo agonist-induced conformational transitions [41].

The above observation was interpreted to suggest that both cholesterol and anionic lipids are essential for nAChR function and that both lipids exert their functional effects by binding to distinct allosteric sites, a view still prevalent in the literature. Three subsequent observations, however, suggested that neither lipid/lipid-type is essential for the nAChR to undergo agonist-induced conformational transitions. First, increasing levels of either cholesterol or the anionic lipid, phosphatidic acid (PA), in a PC membrane stabilize an increasing proportion of agonist-responsive nAChRs, although PA is more effective in this regard [42,43]. Second, in the presence of anionic lipids a variety of neutral lipids substitute for cholesterol in supporting nAChR function [13,44,45]. Finally, in the presence of cholesterol a variety of anionic lipids substitute for PA in supporting a functional nAChR. Collectively these observations show that if both cholesterol and anionic lipids influence function by binding to distinct allosteric sites, then the lipid specificities for these sites are low and their occupancies not absolutely required for an agonist-induced response.

There are also intriguing differences in the capacities of anionic lipids to influence the agonist-induced response. For example, PC membranes containing high levels of PA stabilize a large proportion of agonist-responsive nAChRs, while PC membranes containing similar levels of phosphatidylserine (PS) or other anionic lipids do not [46,47,48]. These and other observations [43] suggest that PA has a unique capacity to stabilize an agonist responsive nAChR. One possibility is that the small anionic headgroup allows PA to bind with higher affinity and thus greater occupancy to an allosteric site to promote channel function. Another is that high levels of PA increase the ordering of the surrounding bilayer, possibly in a manner that mimics the ordering observed in the presence of cholesterol [48]. High levels (40 mol%) of PA in a PC membrane may be particularly effective at stabilizing a functional nAChR because PA exhibits both the required anionic headgroup charge and an ability to influence bulk membrane physical properties in a manner that supports agonist-induced conformational transitions. Further supporting a role for bulk membrane physical properties in nAChR function, hydrophobically thick PC membranes promote conformational transitions even in the absence of cholesterol and anionic lipids [49].

#### 1.2.2. Sites of Lipid Action at the nAChR

Early biophysical and computational studies suggested that cholesterol, and possibly other lipids, bind to both annular and non-annular sites on the nAChR to influence function [50,51,52]. Annular sites are those located at the periphery of the TMD in rapid exchange with lipids in the bulk membrane environment, while non-annular sites are those buried between TMD α-helices that are shielded from bulk membrane lipids [53]. In contrast to the plethora of annular lipid sites observed in nAChR structures (discussed below), none of the nAChR structures solved to date exhibits density attributed to buried non-annular lipids. Both the abundance of observed annular lipid sites, which should be more mobile than non-annular lipids, and the absence of observed buried non-annular lipids argue against the existence of functional non-annular lipid binding to the nAChR. On the other hand, structures of the nAChR and other pLGICs reveal density due to annular lipids, but with the acyl chains extending in between TMD 𝛼-helices (see below). In these cases, the distinction between annular and non-annular lipid binding is blurred.

The first direct structural evidence for annular lipid binding to the nAChR was obtained from cryo-electron microscopy (cryo-EM) structures of the detergent-solubilized human neuronal 𝛼4𝛽2 nAChR (both 𝛼4_3_𝛽2_2_ and 𝛼4_2_𝛽2_3_ stoichiometries) and the azolectin nanodisc-reconstituted human neuronal 𝛼3𝛽4 nAChR, the two sets of structures solved in the presence of the water-soluble cholesterol analog, cholesterol hemisuccinate (CHS) [54,55]. Each structure exhibits regions of electron density at the periphery of the TMD that was modeled as cholesterol (Figure 2). The bound cholesterol, located in the cytoplasmic leaflet at both the M4–M1 and the M4–M3 interfaces of each subunit, is close to residues covalently labeled in the *Torpedo* nAChR by a photoactivatable cholesterol probe [56]. Notably, the electron density attributed to cholesterol disappears when the cryo-EM samples are prepared in the absence of CHS.

Annular cholesterol sites are observed in cryo-EM structures of the azolectin nanodisc-reconstituted *Torpedo* nAChR, which were solved using receptors purified from native *Torpedo* membranes [57]. Three *endogenous* cholesterol sites, deemed high affinity, are observed bound to an intracellular leaflet hydrophobic pocket framed by M4, M3 and MX on the principal face of the two 𝛼 subunits and the single 𝛽 subunit. In all three cases, the planar sterol ring is sandwiched between a valine and an arginine on M3 (e.g., 𝛼V294 and 𝛼R301 with 𝛼R301 projecting towards the hydroxyl of cholesterol), and a valine/isoleucine and phenylalanine on MX (e.g., 𝛼V312 and 𝛼F316). Each of the observed cholesterol binding poses overlaps with, but is distinct from, those observed at the M4–M3 interface in the neuronal 𝛼4𝛽2 and 𝛼3𝛽4 nAChRs—the distinct poses could reflect improved modeling due to the higher resolution of the *Torpedo* structures (2.6 Å for the highest resolution *Torpedo* structure versus 3.3 and 3.5 Å for the 𝛼3𝛽4 and 𝛼4𝛽2 structures, respectively) or different binding of CHS versus cholesterol. Additional cholesterol sites are observed in structures solved in the presence of exogenously added cholesterol. These extra sites, deemed low affinity, are found in the extracellular leaflet where they frame either side of M4.

It is notable that the cholesterol sites observed in the 𝛼3𝛽4, 𝛼4𝛽2 and *Torpedo* nAChR structures overlap with regions of low electron density in cryo-EM images recorded from native *Torpedo* post-synaptic membranes, with the low-density regions attributed to bound cholesterol [58,59,60]. The bound cholesterol is observed at both inner and outer leaflet transmembrane sites. Interestingly, the presence of cholesterol stabilizes a “splayed-apart” arrangement of the M1–M3–M4 α-helices in the outer leaflet of the bilayer, with this arrangement postulated to create space for the pore-lining M2 α-helices to move during gating [59,60]. Cholesterol-interacting regions become more extensive, thus leading to the formation of microdomains in areas bridging adjacent receptors, particularly in the vicinity of the disulfide linkage between δ-δ dimers of neighboring nAChRs.

Annular sites for phospholipids are also observed in each of the *Torpedo* nAChR structures solved to date, including a conserved inner leaflet site adjacent to, but in some cases overlapping with the high affinity cholesterol sites noted above [57,61,62]. In most cases, the phosphate of the modeled phosphatidylcholine (PC) is sandwiched between two positively charged residues, a conserved arginine located just after the M3 𝛼-helix from the principal subunit and a lysine, arginine, or histidine from the complimentary M4 𝛼-helix (e.g., 𝛼R301 and 𝛾K449 at the 𝛼-𝛾 site; Figure 3). The conserved arginine is positioned by a conserved coordinating tryptophan on the M4 𝛼-helix (e.g., 𝛼W399) and a M3–MX loop histidine (e.g., 𝛼H306), both on the principal face of the lipid binding site. An aromatic side chain from the complimentary M1 𝛼-helix (e.g., 𝛾F242) also forms a stacking interaction with one acyl chain. Note that the binding pose of the choline moiety of the headgroup varies from subunit to subunit and from structures to structure likely because there are no specific coordinating interactions. Furthermore, PC or PA bind quickly to this motif and remain bound for the duration of all trajectories in molecular dynamics (MD) simulations [62]. This site is connected to a salt bridge between M4 and the back of the M2 𝛼-helix, previously shown to be important in channel gating in the human adult muscle nAChR [63]. The five noted residues may constitute a signature motif for a functionally important phospholipid binding site that could have a particularly high affinity for PA.

In the extracellular leaflet, the annular phospholipids typically bind to a shallow cavity formed by a positively charged residue on M3 (e.g., 𝛼K276), the M2–M3 loop and the Cys-loop from the principal subunit along with M1 from the complementary subunit. The bound phospholipid is located between, but in some cases overlaps with the two extracellular leaflet cholesterol sites. It is notable that the outermost M4 𝛼-helix from 𝛼_𝛿_, and to a lesser extent from 𝛼_𝛾_, tilts away from the rest of the TMD in agonist bound structures. The outward tilt of M4 allows the acyl chain of an outer leaflet phospholipid to enter the void between the end of M4 and the rest of the TMD where it contacts the strictly conserved Cys-loop FPF motif [62]. The outward tilt of M4 allows the inhibitor d-tubocurarine to bind in the same cavity [57]. It has been proposed that dynamic movements of M4 underlie both the uncoupling of binding and gating that occurs with the *Torpedo* nAChR reconstituted into PC membranes lacking cholesterol and anionic lipids [46]. Dynamic movements of M4 leading to altered lipid and/or allosteric modulator binding is a common theme observed in several pLGIC structures, as discussed below.

It is intriguing that although the actual number of bound lipids generally increases with the resolution of the solved structure, several bound lipids are observed in each of the nine *Torpedo* structures solved to date. In contrast, structures of the neuronal 𝛼7 nAChR exhibit either no or only a few bound lipids. Specifically, structures of the 𝛼7 nAChR reconstituted into similar azolectin nanodiscs do not exhibit any bound lipids even though they were solved at relatively high resolutions (2.7 Å to 3.6 Å) [64]. In a second study, diffuse density was observed at the periphery of the TMD of 𝛼7 nAChR structures solved in detergent, with one region of electron density in each subunit modeled as cholesterol (Figure 2) [65]. One speculative interpretation derived from this observation is that the membrane facing surface of the *Torpedo* nAChR TMD is more amenable to lipid binding than the membrane exposed surface of the 𝛼7 nAChR. An increased propensity to bind lipids could underlie the exquisite functional sensitivity of the *Torpedo* nAChR to its membrane environment.

Another striking feature of the various *Torpedo* structures is that, while there is conservation of phospholipid binding sites, which hints at a role for such sites in channel function, the bound cholesterol observed in structures reported by Rahman et al. [57] are not observed in all *Torpedo* structures, particularly those reported by Zarkadas et al. [62]. The absence of bound cholesterol in the latter structures likely reflects the fact that Zarkadas et al. washed the detergent-solubilized nAChR extensively with detergent-solubilized soybean azolectin during purification leading to cholesterol–phospholipid exchange. The Rahman et al. and Zarkadas et al. *Torpedo* structures show, not surprisingly, that the protocol used to prepare a pLGIC for cryo-EM imaging can dramatically alter the observed lipid binding. Comparison of the structures also shows that phospholipids and cholesterol bind to overlapping annular sites, thus giving credence to the hypothesis that the low lipid specificity of the *Torpedo* nAChR results, at least in part, from the binding of different lipids to overlapping sites, albeit with different affinities/occupancies and different efficacies for stabilizing an agonist-responsive nAChR [13,46].

The ensemble of solved nAChR structures also highlight limitations in how structural data can inform our understanding of lipid–nAChR interactions. As noted above, the pattern of bound lipids differs depending on the protocols used to prepare the cryo-EM samples. There may also be subtle differences in lipid binding that result from the different scaffolding proteins used to encapsulate the reconstituted nAChR in a lipid nanodisc. For example, lipid binding in the Zarkadas et al. structures is suggestive of distinct lipid sites, while lipid binding in some of the Rahman et al. structures approaches a continuous annular belt between the M4 𝛼-helices from adjacent subunits (Figure 2). Zarkadas et al. imaged the nAChR embedded in nanodiscs formed using the circular scaffolding protein, MSP2N2, with two molecules of the scaffolding protein surrounding a lipid bilayer with a fixed diameter of 15–17 nm, while Rahman et al. imaged the nAChR in nanodiscs formed using the scaffolding protein, saposin A. The more globular saposin A monomers assemble in different stoichiometries to form nanodiscs with different diameters, depending on the size of the encapsulated membrane protein. It has been suggested that saposin A encapsulates membrane proteins with a minimal number of lipids trapped between the membrane protein and each saposin A monomer [66]—an observation supported by recent structures of the 5-HT_3_R (see below). The additional lipids observed in structures solved by Rahman et al. could partly reflect a tighter saposin A nanodisc that prevents the diffusion of encapsulated lipids within the nanodisc, thus allowing the detection of both high affinity allosteric and lower affinity annular sites.

Finally, it is significant that the virtually superimposable structures, solved by Zarkadas et al. and Rahman et al., in the presence of the agonist carbamylcholine (Carb) were attributed to different physiological states. Prolonged exposure to Carb should lead to a stable desensitized conformation that binds agonist with high affinity but does not flux cations across the membrane, a finding consistent with functional assays performed on the nanodisc reconstituted nAChR. Backbone restrained MD simulations suggest that the pore of the agonist-bound nAChR is hydrated and likely conductive for cations, a finding inconsistent with a non-conductive desensitized conformation [62]. In unrestrained MD simulations, the pore collapses to a non-conductive conformation like that observed in the resting state. Diffuse density, however, was observed in the pore of the Carb and nicotine bound structures. When this density was modeled as lipids, the conformation of the nAChR was stable in backbone unrestrained MD simulations, with the dynamic lipid blocking hydration of the pore and, thus, cation flux.

The simplest interpretation of the MD simulations is that lipid, possibly from lipid vesicles or empty lipid nanodiscs that are destroyed during sample vitrification, lodges in the open pore in the presence of Carb and traps the nAChR in a transient conformation along the reaction coordinate between an open or pre-open state and the desensitized state. A second speculative interpretation is that the structures represent a true desensitized state, but that the blockage of cation-flux upon desensitization results from the diffusion of lipids from the surrounding bilayer into the pore. Note that lipids have been observed bound to the pores of other ion channels, such as the bacterial mechanosensitive channel, MscS, and have been proposed to play a role in mechanosensitive channel gating [7,67]. Lipid headgroup density has also been observed penetrating into the wide open pore of the prokaryotic pLGIC, DeCLIC [19]. Regardless, the lack of clarity regarding the physiological state of the agonist bound structures highlights the ongoing struggle to definitively assign solved structures to physiologically relevant conformations—particularly for lipid-sensitive ion channels. The limitations in our ability to definitively assign structures to physiological states impacts our ability to elucidate state-dependent lipid–nAChR interactions and, thus, fully understand the mechanisms by which lipids influence nAChR function.

#### 1.2.3. Mechanisms of Lipid Action at the nAChR

Functional assays show that cholesterol and anionic lipids are important to nAChR function and that lipids influence the magnitude of the agonist-induced response mainly by interacting preferentially with and, thus, preferentially stabilizing different proportions of activatable (resting) versus non-activatable (desensitized or uncoupled) conformations [46,68]. Lipids can also interact with transition states to influence the rates of conformational transitions [49]. Despite the available structures revealing both cholesterol and phospholipid binding sites on the nAChR, it remains equivocal as to whether lipids preferentially stabilize different conformations by binding to allosteric sites, by altering bulk membrane physical properties that in turn preferentially stabilize different conformations, or by a combination of both. Although the observation that cholesterol and phospholipids bind to mutually overlapping sites gives credence to the hypothesis that the low specificity of the *Torpedo* nAChR for different lipids may result from the binding of different lipids to overlapping sites, albeit with different affinities/occupancies and different efficacies for stabilizing an agonist-responsive nAChR [13,46], a definitive functional role for lipid binding in nAChR function remains to be equivocally established. In this context it is notable that both the apo and the agonist-bound *Torpedo* structures solved in the presence and absence of bound cholesterol are virtually superimposable suggesting that cholesterol binding does not alter nAChR structure in a manner that influences nAChR function. The mechanisms by which lipid binding and/or membrane physical properties interact preferentially with and, thus, stabilize one conformation over another remains a central unanswered question underlying the mechanisms of nAChR–lipid interactions.

Both the *Torpedo* and 𝛼7 nAChR structures solved in the presence and absence of ligands reveal structural changes in the lipid-exposed TMD surface, which could underlie conformation-specific interactions with lipids and/or bulk membrane properties. The *Torpedo* and 𝛼7 nAChR structures solved in different conformations provide a starting point for understanding conformationally specific nAChR–lipid interactions. In addition, the lipid-dependent uncoupled nAChR observed in PC membranes lacking cholesterol and anionic lipids exhibits enhanced rates of peptide hydrogen exchange relative to the resting and desensitized conformations suggesting that a region(s) of the polypeptide backbone that is buried from the aqueous solvent in both the resting and desensitized states becomes exposed to solvent in the uncoupled state [41]. Given the importance of the interface between the ECD and TMD in “coupling” agonist binding to channel gating, it was proposed that lipid-dependent uncoupling results from weakened interactions and, thus, a physical separation at the ECD–TMD interface. Further structural insight into the uncoupled state should shed light on the mechanisms by which lipids alter channel gating.

Lipids and/or bulk membrane physical properties could influence the coupling of binding and gating via the lipid exposed M4 𝛼-helix, which packs against the adjacent TMD 𝛼-helices, M1 and M3. Altered M4–M1/M3 interactions have long been thought to play a role in translating nAChR–lipid interactions into altered channel function, a hypothesis supported by the observation that mutations that influence the strengths of M4–M1/M3 interactions modulate the functional sensitivity interface of the prokaryotic pLGIC, ELIC, to lipids (see below) [69]. Lipid-dependent alterations in the packing of M4 against M1/M3 may modify interactions between the M4 C terminus and the Cys-loop, a structure at the ECD–TMD interface that is important in channel gating [22]. Interactions between M4 and the Cys-loop (i.e., the 𝛽6–𝛽7 loop) are critical for folding and function of the prokaryotic pLGIC, GLIC and the 𝛼7 nAChR [64,70]. On the other hand, M4 C-terminal deletions have little effect on function in the human adult and *Torpedo* nAChR, suggesting that M4–Cys-loop interactions are not critical for function [62,71]. Although altered M4–Cys-loop interactions could underlie altered coupling of binding and gating in some pLGICs, they appear to be unimportant in others. The observation that M4 tilts away from the rest of the TMD upon Carb binding to the *Torpedo* nAChR leading to altered lipid binding lends support to the idea that M4 acts as a lipid sensor that translates changes in the surrounding lipid environment into altered nAChR function.

### 1.3. Serotonin Receptor (5-HT_3_R)

#### 1.3.1. Functional Sensitivity of the 5-HT_3_R to Lipids

Although several studies suggest that both the structure and function of 5-HT_3_Rs are sensitive to lipids, the specific effects of lipids on agonist-induced 5-HT_3_R gating remain to be characterized [13]. On the other hand, alanine and other substitutions show that the putative M4 lipid sensor influences mouse 5-HT_3A_R function, possibly in a lipid-dependent manner [72,73]. Deletion or mutation of the C-terminal alanine residue leads to a decrease in expression of 5-HT_3A_Rs, consistent with a role for the M4 C-terminus in folding/trafficking [74]. Unlike the *Torpedo* nAChR, interactions between the M4 C-terminus and the ECD appear to be critical in the 5-HT_3A_R. The variabilities observed in 5-HT_3_R structures solved in detergent with and without added lipids versus in lipid nanodiscs, as discussed below, provides the most compelling evidence that lipids play a significant role in 5-HT_3_R function.

#### 1.3.2. Sites of Lipid Action at the 5-HT_3_R

Several studies over the past ~10 years have reported structures of the detergent-solubilized homomeric 5-HT_3A_R in multiple conformational states with and without added lipids, including numerous structures in complex with a class of drugs, setrons, that are used to manage vomiting associated with both radio and chemotherapies [75,76,77]. Even the highest resolution structure of the detergent-solubilized 5-HT_3A_R solved in the presence of added lipids (2.8 Å resolution), however, lacks density at the periphery of the TMD that could be confidently modeled as lipid [77]. In contrast, structures of the 5-HT_3A_R reconstituted into saposin A lipid nanodiscs reveal the presence of bound annular lipids [78]. In both apo and serotonin-bound open states, density attributed to cholesterol is observed in the extracellular leaflet in a shallow groove formed by M1, M4 and the Cys loop, a position close to that seen with low affinity cholesterol binding to the nAChR. It was suggested that cholesterol stabilizes tighter interactions at the ECD–TMD interface, thus leading to a conformation where agonist-binding is coupled to channel gating. Additional densities attributed to phospholipids were also seen in the serotonin-bound structures in the extracellular leaflet at an inter-subunit site between M1 and M3 from the complementary and principal subunits, respectively, with the lipid headgroup projecting towards the pore-lining M2 𝛼-helices where they make additional contacts with the M2–M3 linker. Inter-subunit density was not observed in apo structures where the inter-subunit cavities are smaller [78].

MD simulations using the original 3.5 Å X-ray structure of the 5-HT_3A_R as a template have explored 5-HT_3A_R–lipid interactions. In a heroic 15–20 µs atomistic simulation of the 5-HT_3A_R imbedded in a bilayer composed of stearoyl-docosahexaenoyl PC, palmitoyl-oleoyl PC and cholesterol, stearoyl-docosahexaenoyl PC adhered more compactly to the TMD ultimately leading to clustering around the TMD periphery [79]. One MD simulation detected simultaneous hydrogen-bonding between a bound phospholipid headgroup and both the M2-M3 loop and the Cys loop, bridging interactions that could promote functional coupling between the ECD and TMD [80]. Dynamic movements of the C-terminal half of M4 led to gaps in the extracellular leaflet at the M4–M1/M3 interface, these gaps can be filled by both phospholipid acyl chains and cholesterol, as is observed structurally with the *Torpedo* nAChR. As noted, dynamic movements of M4 leading to altered interactions with lipids and other allosteric modulators appear to be a common theme of all pLGICs.

One notable limitation of the MD simulations is the conformational ambiguity of the 5-HT_3A_R structures solved in different laboratories. For example, the most recent nanodisc-reconstituted 5-HT3_A_R open structure solved in the presence of serotonin exhibits larger displacements of the M1, M3, M4 and MX 𝛼-helices leading to a larger pore diameter than observed in the serotonin-bound open structures solved in detergent either with or without added lipid. The two reported open structures solved in detergent with or without added lipid result from symmetric movements of the five subunits, while the nanodisc-reconstituted open structure results from asymmetric subunit movements. Further complicating the interpretation of these conformations, MD simulations provide conflicting evidence as to whether the pores in the open states are hydrated and thus have the capacity to flux cations. Although the observed conformational differences emphasize that structural sensitivity of the 5-HT_3__A_R to lipids, the inability to unequivocally assign solved structures to defined physiological states limits our ability to define conformationally specific lipid binding. It is also intriguing that saposin A scaffolding protein in the nanodisc-reconstituted symmetric apo 5-HT_3A_R structure interacts tightly with the outermost TMD 𝛼-helices, including M4 and MX (see Supplemental Figure 1f in [78]). It will be interesting to probe whether direct interactions between saposin A and the 5-HT_3A_R alter the structure of the TMD in a manner that influences lipid binding.

#### 1.3.3. Mechanisms of Lipid Action at the 5-HT_3_R

Although the noted differences in 5-HT_3_R structure in different lipid environments provide compelling evidence for a functional sensitivity to lipids, there is currently insufficient data regarding both the effects of lipids on 5-HT_3_R function and the modes of lipid binding to develop detailed models of 5-HT_3_R–lipid interactions. Despite this, both structural and MD simulations suggest that the lipid-exposed M4 𝛼-helix moves relative to M1/M3 during channel gating [27] to alter the shape of a cavity at the M4–M1/M3 interface, which is the proposed site for binding of the lipophilic allosteric potentiator, trans-3-(4-methoxyphenyl)-N-(pentan-3-yl)acrylamide [27]. This cavity is one of the sites proposed for cholesterol and neurosteroid binding to the nAChR. Furthermore, dynamic movements of M4 have been proposed to underlie lipid-dependent uncoupling of binding and gating in the nAChR. The cavity between M4 and M1/M3 may be an allosteric site for lipids and other lipophilic compounds that is conserved in all pLGICs.

### 1.4. GABA_A_ Receptors

#### 1.4.1. Functional Sensitivity of the GABA_A_R to Lipids

Limited functional studies suggest that both cholesterol and the anionic lipid, phosphatidylserine (PS), modulate GABA_A_R function. Specifically, incubating both GABA_A_R containing synaptosomal membranes and detergent-solubilized native GABA_A_Rs from rat cerebral cortex with increasing concentrations of PS enhances binding of the benzodiazepine, flunitrazepam [81]. Methyl-β-cyclodextrin-induced depletion of cholesterol levels in the cell membranes of rat hippocampal neurons expressing GABA_A_Rs diminished the magnitude of the agonist-induced response, an effect that was not reversed by enrichment with the cholesterol stereoisomer, epicholesterol [82]. Based on the observation that neurosteroids are less effective at modulating GABA_A_R function in cholesterol-enriched membranes, it was concluded that cholesterol and neurosteroids bind to overlapping sites [83]. Similar to the nAChR, optimal levels of cholesterol in native membranes are required, with both increasing or decreasing cholesterol levels away from endogenous levels diminishing the agonist-induced response [42,83]. Membrane vesicle size, which influences membrane curvature, modulates flunitrazepam binding implicating a role for bulk membrane physical properties in GABA_A_R function [84].

#### 1.4.2. Sites of Lipid Action at the GABA_A_R

A structure of the detergent solubilized α1β2γ2 GABA_A_R solved in the presence of CHS at a resolution of 3.9 Å was the first GABA_A_R structure to identify bound lipids [85]. Regions of electron density attributed to twelve molecules of CHS were observed predominantly at annular sites, although density at subunit interfaces penetrates deeper into the TMD. A subsequent structure of the agonist-responsive human synaptic α1β3γ2 GABA_A_R solved in MSP2N2 lipid nanodiscs at higher 3.2 Å resolution, however, suggested that the initial α1β2γ2 GABA_A_R structure does not represent a true physiological state [86]. Although no cholesterol was observed bound to the nanodisc-reconstituted α1β3γ2 GABA_A_R, electron density at the M4–M1 interface in the extracellular leaflet was modeled as PC. Two molecules of 4,5-bis phosphate (PIP_2_) were also observed bound between M4 and M3 in the intracellular leaflet of the α1 subunit, with the two PIP_2_ binding sites subsequently confirmed in a full length α1β3γ2 GABA_A_R structure solved at 2.7 Å resolution [17]. The two bound PIP_2_ molecules are notable because they represent the only phospholipids bound to a eukaryotic pLGIC structure that have been unambiguously identified. In addition, the PIP_2_ headgroup is highly coordinated by positively charged residues extending from the intracellular loop following M3, the base of M4, and the M1–M2 loop, thus suggesting an important role in GABA_A_R function (Figure 4) [86].

Note that, despite the lack of observed cholesterol binding to the α1β3γ2 GABA_A_R, crystal structures of chimeras with GABA_A_R TMDs (a homomeric ELIC ECD–α1 GABA_A_R TMD, a GLIC ECD–α1 GABA_A_R TMD and a β3 GABA_A_R ECD–α5 GABA_A_R TMD) exhibit both inhibiting and potentiating neurosteroids bound to intracellular leaflet sites between M4 and M3 and between M3 from the principal subunit and M1/M4 from the complementary subunit, respectively [87,88,89]. Both these sites overlap with binding sites for CHS in the original α1β2γ2 structure, with the functional sensitivity of these sites to neurosteroids supported by both mutagenesis and affinity labeling [90,91].

The highest resolution structure (2.8 Å) within a set of human α1β3 GABA_A_R structures solved in bovine brain lipid nanodiscs also exhibit numerous regions of electron density at the periphery of the TMD in both leaflets of the bilayer, although the regions of electron density are too small to provide detailed insight into the modes of lipid binding [92]. Of note, one region of electron density penetrates an inter-subunit site between the principal α1 and complementary β3 subunits, as observed above in the original α1β2γ2 GABA_A_R structure. Allosteric modulators, such as phenobarbital, etomidate and propofol, bind to an inter-subunit cavity in α1β2γ2 GABA_A_R structures, although not at the α1–β2 interface where lipid binding is observed [93]. Lipids may bind to an inter-subunit allosteric site in the TMDs of many GABA_A_Rs.

Docking studies further suggest the existence of cholesterol binding sites on GABA_A_R structures, including one site in the extracellular leaflet at the inter-subunit interface [94]. The inability of epicholesterol to substitute for cholesterol in supporting GABA_A_R function suggests the existence of specific cholesterol regulatory sites, a hypothesis supported by another docking study which proposed that interactions between cholesterol and the β3 homo-pentameric GABA_A_R are not mimicked by the binding of epicholesterol [95].

#### 1.4.3. Mechanisms of Lipid Action at the GABA_A_R

Despite compelling biochemical and structural data suggesting that GABA_A_R function is sensitive to lipids, the mechanisms by which lipids modulate GABA_A_R function remain unclear. One intriguing MD simulation showed that the binding of positive allosteric modulators to the α1β2γ2 GABA_A_R attenuates local motions within the TMD, whereas an allosteric antagonist enhances TMD motions leading to an altered structure of the ECD and the dissociation of GABA. Lipids and lipophilic drugs may influence function by stabilizing the TMD structure in an optimal conformation for interactions with the ECD [93,96], as suggested for the nAChR [41]. The observed PIP_2_ binding sites on the α1β3γ2 GABA_A_R are particularly intriguing sites for lipid action, although the minimal functional data obtained to date suggest that PIP_2_ binding serves a role in receptor trafficking, as opposed to channel gating [86].

### 1.5. Glycine Receptors

#### 1.5.1. Functional Sensitivity of the GlyR to Lipids

GlyRs purified from rat spinal cord and reconstituted into soybean azolectin membranes are capable of undergoing agonist-induced anion flux [97]. Beyond that, however, no studies have systematically examined the effects of different lipids on GlyR function. The single channel conductance of both homomeric α2 and heteromeric α2β GlyRs are similar in both planar bilayers composed of either polar lipids extracted from brain tissues or a combination of both phosphatidylethanolamine and phosphatidylglycerol [98]. Depletion of cholesterol from HEK cell membranes expressing homomeric α1 or α3 and heteromeric α1β GlyR using methyl-𝛽-cyclodextrin did not affect the maximal glycine-induced current; although it did inhibit the potentiation of GlyR currents by the cannabinoid; tetrahydrocannabinol [99]. Interestingly; M4-swapped chimeric constructs of the homologous glycine receptor α1 and α3 subunits demonstrate that subunit-specific agonist efficacy is driven in large part by M4 and residues at the M4-lipid interface [100]. M4 is crucial for trafficking of the GlyR to the cell surface [101]. M4–M1/M3 interactions play a critical role in channel folding and function [101,102]. The structural variability of GlyR structures solved in detergents versus different lipid nanodiscs highlight a conformational sensitivity of the GlyR to lipids, as discussed below.

#### 1.5.2. Sites of Lipid Action at the GlyR

Several studies have reported structures of GlyRs with densities attributed to bound lipids. The structure of an apo closed state of the zebrafish α1 GlyR reconstituted into soybean azolectin nanodiscs exhibited three densities in each subunit attributed to bound phospholipids, with two lipids binding to the extracellular leaflet at the M1–M4 and M4–M3 interfaces and one lipid binding to the inner leaflet at the M4–M3 interface [103]. No coordinating interactions were observed between the α1 GlyR and the bound lipid headgroups, suggesting that these phospholipid sites, as in other pLGICs, have low lipid specificity. Interestingly, the inner leaflet phospholipid binds predominantly to the M4–M3 interface of the principal subunit at a site that overlaps with the high affinity cholesterol sites observed in the nAChR, but not with the conserved intracellular phospholipid site on the complementary subunit of the nAChR. The lack of an ordered MX α-helix may shape distinct phospholipid binding to the GlyR.

A second study reported structures of both heteromeric and homomeric GlyRs detergent-solubilized from porcine spinal cord and brain stem [18], with the highest resolution heteromeric 𝛼1𝛽 GlyR structure (2.7 Å) exhibiting 45 regions of electron density surrounding the TMD. Unfortunately, these regions of density were too small to accurately model bound lipids. Another study of the zebrafish α1 GlyR reconstituted into both SMA and MSP1E3D1 nanodiscs yielded numerous structures ranging in resolution from 2.9 to 4.0 Å, with similar densities at the periphery of the TMD corresponding to between 47 and 65 lipid fragments [104]. No densities attributable to bound lipids were observed in the structures solved in SMA nanodiscs, despite the structures being solved at similar or higher resolutions. The diffuse nature of the observed lipid densities suggests that they reflect nonspecific annular bound lipids rather than lipids bound with high affinity to allosteric sites (Figure 5). No conserved interactions between phospholipids and the GlyR have been detected, suggesting that lipid effects on function, if they exist, likely occur through packing or bulk membrane properties rather than through specific modulatory sites.

Of particular note, an early 𝛼1 GlyR structure solved in detergent exhibited a super-open conformation with a pore diameter ~8.8 Å [26,105]. Cryo-EM data sets obtained for the 𝛼1 GlyR in MSP1E3D1 nanodiscs captured numerous conformations including a super-open conformation with a pore diameter of ~7 Å that was deemed non-physiological as the pore diameter is large enough to conduct the impermeant organic anion, isethionate [104]. The data set obtained using MSP1E3D1 nanodiscs also captured a smaller open pore conformation with a diameter of ~5.6 Å, which is consistent with the open pore diameter of 5.3 Å predicted from single channel conductance measurements. Significantly, extracting the 𝛼1 GlyR directly from the insect cell membranes in which it was expressed increased the proportion of receptors in the physiologically relevant open state. These variable structures highlight the exquisite conformational sensitivity of the 𝛼1 GlyR to lipids.

Conformationally specific lipid binding to the GlyR has been further explored in coarse-grained MD simulations using a homology model of the homomeric human α1 GlyR closed structure and the homomeric zebrafish α1 GlyR super-open structure [106]. One intriguing finding of this study was that cholesterol binds to an inter-subunit site at a location similar to the site of ivermectin binding to both the *C. elegans* glutamate-activated chloride channel (GluCl) and the α3 GlyR [107,108]. Cholesterol binding was observed in simulations of the super-open state, whereas in the closed state cholesterol remained at the periphery of the TMD. These experiments are significant because they demonstrate that it is possible to use MD simulations to investigate conformation-specific lipid binding. Given that the super open conformation structures used in this study was subsequently deemed non-physiological, the data further highlight the need for definitive structure to state assignments to define pLGIC–lipid interactions.

#### 1.5.3. Mechanisms of Lipid Action at the GlyR

As with the 5-HT_3_R, there is currently insufficient structural and functional data to develop detailed models regarding the mechanisms by which lipids modulate GlyR function. Despite this, one intriguing feature of both GABA_A_Rs and GlyRs is that they exhibit an extensive network of interacting aromatic residues at the M4–M1/M3 interface. In the prokaryotic pLGICs, GLIC and ELIC, a similar network of interacting residues has a dramatic effect on their functional sensitivities to lipids [69], as discussed below.

### 1.6. Prokaryotic pLGICs

#### 1.6.1. Functional Sensitivity of Prokaryotic pLGICs to Lipids

The two prokaryotic pLGICs, GLIC and ELIC, have become attractive models for probing the mechanisms underlying pLGIC–lipid interactions. Although the effects of lipids on GLIC and ELIC function have not been characterized extensively, GLIC retains the ability to undergo agonist-induced channel gating when reconstituted in a minimal PC membrane, while ELIC does not [69,109]. The ability of GLIC to function in a PC membrane was attributed to the presence of an extensive network of interacting aromatic residues at the M4–M1/M3 interface, which stabilizes the TMD structure, likely rendering it less malleable and thus less sensitive to changes in the surrounding lipid environment [101]. In contrast, ELIC has fewer aromatic residues at this interface, which may sterically prevent effective M4–M1/M3 interactions leading to a more malleable TMD structure that requires an optimal lipid environment for channel function [110]. Consistent with this hypothesis, transplanting the GLIC aromatic-interacting network into ELIC restores its ability to undergo agonist-induced channel gating in minimal PC membranes [69]. In addition, reconstituting wild-type ELIC into PC membranes containing either of the anionic lipids, phosphatidylglycerol (PG) or cardiolipin, restores a robust agonist-induced response [32,111,112].

Cholesterol and the fatty acid, docosahexaenoic acid (DHA), influence both GLIC and ELIC function, albeit with different phenotypic effects. Increasing levels of cholesterol in a reconstituted membrane either enhances or reduces the rates of GLIC and ELIC desensitization, respectively. ELIC desensitization is slowed even further in the presence of the anionic lipid, PG (see below) [32,111,113]. The fatty acid DHA increases the rates of GLIC desensitization [114], while treatment of ELIC with DHA leads to a reduction in the magnitude of the agonist-induced peak current, but does not alter the EC_50_ for channel gating [115].

#### 1.6.2. Sites of Lipid Action at Prokaryotic pLGICs

The “apparently open” crystal structure of GLIC published in 2009 was the first pLGIC structure to detect bound lipids at the periphery of the TMD. One phospholipid was modeled in the extracellular leaflet at the interface between M4 and M1. Two phospholipids were modeled in the intracellular leaflet at the interfaces between both M4 and M3 and M4 and M1, although the latter lipid extends across the inter-subunit interface [116]. Surprisingly, no exogenous lipids were added during either purification or crystallization suggesting that the observed bound lipids are endogenous to the *E. coli* membranes in which GLIC was expressed. The extracellular leaflet lipid overlaps with both a phospholipid site on the α1β3γ2 GABA_A_R and a low affinity cholesterol site on the *Torpedo* nAChR [57,86]. This site is close to a proposed neurosteroid binding site on the α1β3 GABA_A_Rs [91] and overlaps with a cholesterol site identified by photoaffinity labeling on GLIC [117]. The intracellular lipids overlap broadly with proposed neurosteroid and lipid binding sites in a variety of pLGICs.

The extracellular leaflet site in GLIC is of particular interest because it bridges M4 and the 𝛽6–𝛽7 (Cys) loop, with alanine mutations of the bridging residues either impairing function or eliminating functional GLIC expression [70]. This endogenous lipid is not observed in resting and “locally closed” conformations of GLIC, consistent with the bridging lipid stabilizing the open conformation. The inhibitory drugs, propofol and desflurane, bind to an overlapping site, with the allosteric modulators eliminating or altering lipid binding, as discussed elsewhere [118], while DHA, which enhances GLIC desensitization, binds to a site adjacent to the bound PC [114]. Electron paramagnetic resonance suggests that the oxygen accessibility of residues in the lipid binding pocket is enhanced when GLIC transitions from the resting to desensitized states [119,120]. Considerable data thus suggest that lipid binding to this site plays a modulatory role in GLIC function.

Although bound lipids were not initially observed in the original ELIC crystal structures, a relatively high resolution crystal structure detected the binding of phosphatidylethanolamine (PE) to one subunit in the intracellular leaflet at the interface between M4 and M1 [32], a site shown by mass spectrometry to have higher affinity for the anionic lipid, PG [111]. A subsequent cryo-EM structure solved using ELIC extracted from *E. coli* membranes with styrene-maleic acid (SMA) confirmed the presence of PG bound to the same intracellular leaflet site, albeit in each of the five subunits [112]. Cardiolipin was also observed bound to each of the subunits in the extracellular leaflet primarily at an inter-subunit site, although the large lipid with four acyl chains extends both towards the M4-M3 interface of the principal subunit and down through the bilayer into the intracellular leaflet adjacent to the intracellular leaflet PG/PE (Figure 6). The cardiolipin is bound to a site where detergent has been modeled in a previous ELIC structure and is adjacent to a Trp residue that has been implicated in lipid binding [121]. Mutations in several of the cardiolipin binding site residues influence the gating of ELIC. Polyunsaturated fatty acids have also been detected bound to a site at the M4–M1 interface, adjacent to where DHA binds to GLIC [115].

Finally, density, modeled as a PG head group, has been observed at non-annular sites of a recent crystal structure of the prokaryotic pLGIC, DeCLIC [19]. In this wide-open structure, the head-group density is nestled underneath the M2–M3 loop of each subunit and projects in between each of the pore-lining M2 𝛼-helices into the ion channel pore. The observation that bound lipids penetrate the ion channel pore is particularly intriguing given that diffuse electron density has been modeled as lipids bound to the ion channel pore of the *Torpedo* nAChR. The density observed in this region of DeCLIC thus lends credence to the speculative hypothesis that the diffusion of lipids into the pore plays a role in *Torpedo* nAChR desensitization. The functional relevance of lipid binding in DeCLIC, however, requires validation.

#### 1.6.3. Mechanism of Lipid Action at Prokaryotic pLGICs

The improving structural and functional data for prokaryotic pLGICs provide increasing insight into the mechanisms by which lipids influence pLGIC function. Structures of GLIC, ELIC and DeCLIC suggest modulatory lipid sites in the extracellular leaflet, with the structural data highlighting a potential role for lipid binding near the GLIC and ELIC M4 C-terminus in channel function.

The most compelling insight into the mechanisms underlying pLGIC–lipid interactions has been obtained for ELIC. As noted, agonist-induced channel gating is severely impaired when ELIC is reconstituted into PC membranes. Structures of ELIC in this environment suggest that ligand binding leads to conformational changes in the ECD that propagate to the M2–M3 linker but fail to penetrate the remainder of the TMD, an observation consistent with the hypothesis that the physical coupling between the ECD and TMD is lost when the *Torpedo* nAChR is reconstituted into the same PC membranes lacking cholesterol and anionic lipids [41]. Furthermore, a specific functional role has been proposed for PE/PG binding to the intracellular leaflet site. This lipid binding site is shaped by a characteristic W-R-P motif. Both structural and functional data suggest that lipid binding to the intracellular type stabilizes M4 in a kinked conformation to slow the rates of desensitization [32,112]. In the absence of PE/PG binding or with select mutations that eliminate the M4 kink, M4 dynamics likely increase in a manner that leads to the rapid desensitizing phenotype [32]. The links between lipid binding, M4 dynamics and the altered rates of desensitization remain to be fully elucidated.

## 2. Summary and Conclusions

Our understanding of pLGIC–lipid interactions has exploded over the past decade. This understanding is increasingly shaped by new structures of pLGICs solved in different membrane environments. In fact, the plethora of new pLGIC structures solved over the past decade has begun to reveal both the complexities of lipid binding to pLGICs and the conformational transitions that underlie pLGIC function. Combining structural, functional and computational methods will eventually allow researchers to define precisely how lipids interact with pLGICs to preferentially stabilize one conformation over another to modulate pLGIC function. These multidisciplinary studies will also eventually lead to a detailed understanding of the role of pLGIC–lipid interactions in human biology.

Despite the enormous progress, there remain gaps in our knowledge. For most pLGICs, we still do not understand how lipids and bulk membrane properties influence channel gating and desensitization kinetics. We need better functional data on nanodisc-reconstituted pLGICs that will allow us to definitively assign solved structures to conformational states identified by electrophysiological methods. The vast toolbox of biochemical tools available for characterizing the function of the *Torpedo* nAChR should aid in this endeavor [41,62]. We also require a better understanding of how sample purification methods and different nanodisc preparations influence both pLGIC structure and the observed pLGIC–lipid interactions. Although our understanding of the mechanistic underpinnings of pLGIC–lipid interactions still lags behind that of other ion channels, such as inward rectifying potassium channels and mechanosensitive channels, where detailed models describing how signaling lipids and/or bulk membrane properties lead to channel activation have emerged [7], we are certainly at the dawn of a new age where will finally begin to understand the mechanistic underpinnings of pLGIC–lipid interactions.

## Figures and Tables

**Figure 1 biomolecules-12-00814-f001:**
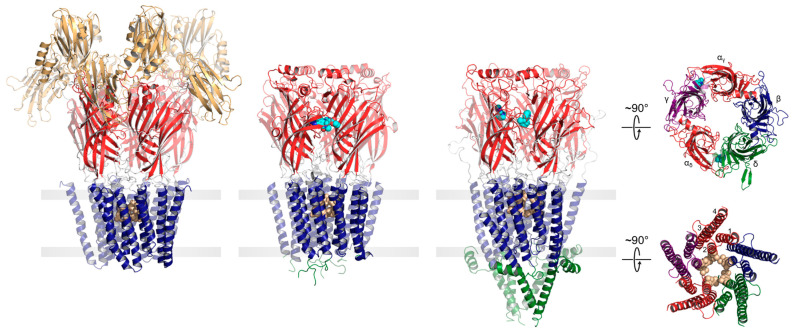
pLGICs display a conserved core architecture with diverse auxiliary features. Side views of the prokaryote DeCLIC (PDB: 6V4S, far left), the human α1β3γ2 GABA_A_R (PDB: 7QNE, middle left), and the *Torpedo* nAChR (PDB: 7QL5, middle right) colored according to domains (NTD, orange; ECD, red; TMD, blue; ICD, green). Bound agonists are presented as cyan spheres at the interfaces between two subunits. In the side views, the principal subunit is on the left and the complementary subunit is on the right. Residues forming the channel gate are presented as tan spheres. Top-down views of the *Torpedo* nAChR ECD (top) and TMD (bottom) are shown on the far right colored according to subunit (α, red; β, blue; γ, purple; δ, green).

**Figure 2 biomolecules-12-00814-f002:**

Lipid binding to both extracellular and intracellular leaflet sites on the nAChR. Side views of the TMD for the cholesterol-bound α3β4 nAChR (PDB: 6PV7, far left) and α7 nAChR (PDB: 7EKI, middle left), phospholipid-bound *Torpedo* nAChR (PDB: 7QL5, middle right), and cholesterol- and phospholipid-bound *Torpedo* nAChR (PDB: 7SMQ, far right) represented as surfaces, with principal and complementary subunits colored in pink and blue, respectively. Bound cholesterol (brown) and phospholipids (yellow) are presented as spheres with oxygen, nitrogen, and phosphorus colored in red, blue, and orange, respectively.

**Figure 3 biomolecules-12-00814-f003:**
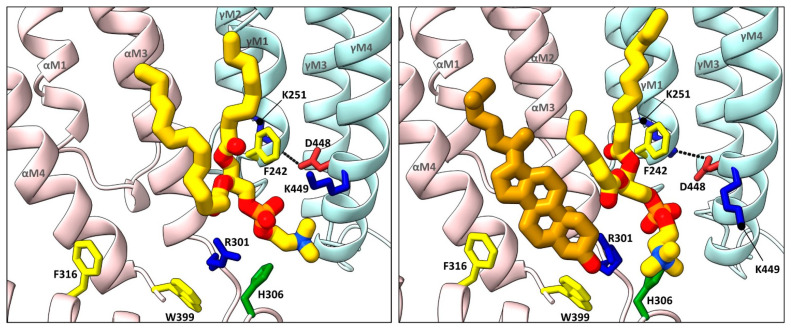
Cholesterol and phospholipids bind to adjacent or overlapping sites in *Torpedo* nAChR structures. Zoomed in views (defined in Figure 2) of *Torpedo* nAChR structures with bound phospholipids (PDB: 7QL5) or both phospholipids and cholesterol (PDB: 7SMQ). Subunits and lipids are colored as in Figure 2, with residues interacting with bound lipids represented as sticks colored according to residue type (non-polar, tan; aromatic, yellow; polar, green; cationic, blue; anionic, red). The M2–M4 salt bridge adjacent to the bound lipids is shown as a dashed line.

**Figure 4 biomolecules-12-00814-f004:**
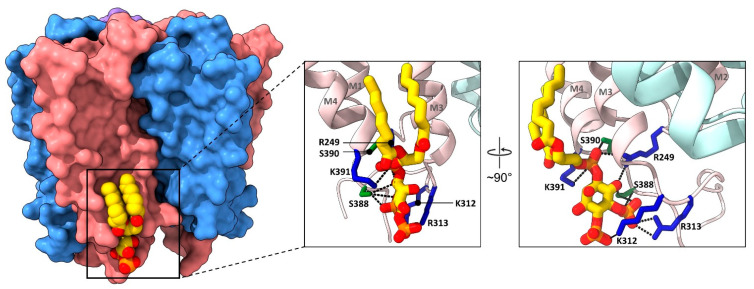
PIP_2_ binds to a highly coordinated site in the α1β3γ2 GABA_A_R. Side view of the α1β3γ2 GABA_A_R TMD (PDB: 7QNE) shown on the left as surface, with a zoomed in view of the boxed region highlighting PIP_2_ and its coordinating residues (colored as residue-type, with positively charged residues colored in blue and neutral hydrogen bonding residues colored in green) represented as sticks on the right. The bound PIP_2_ lipid is colored as in Figure 2, with dashed lines indicating residue-mediated coordination of the head group phosphates. A 90° rotated view is shown on the extreme right to delineate coordinating residues.

**Figure 5 biomolecules-12-00814-f005:**
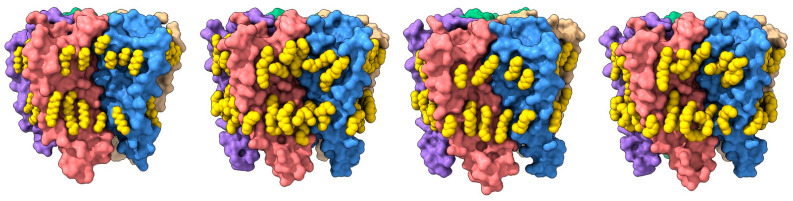
GlyR structures display an annular layer of lipid acyl chains around the perimeter of the TMD. Side views of the detergent-solubilized native porcine GlyR (PDB: 7MLY, far left), GABA-bound SMA-solubilized α1 GlyR (PDB: 6PLU, middle left), taurine-bound SMA-solubilized α1 GlyR in a closed state (PDB: 6PLU, middle right) and taurine-bound SMA-solubilized α1 GlyR in a desensitized state (PDB: 6PLS, far right) with diffuse density corresponding to lipid acyl chains shown as yellow spheres.

**Figure 6 biomolecules-12-00814-f006:**
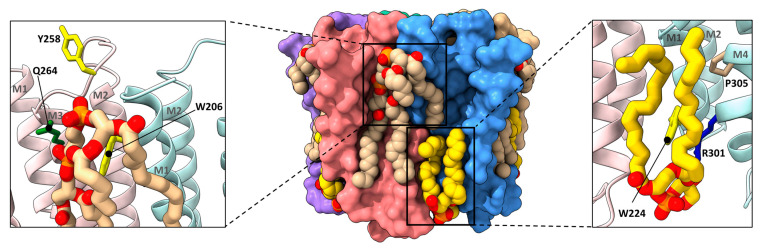
Phospholipid and cardiolipin binding to ELIC. Side view of the ELIC TMD (PDB: 7L6Q) represented as surface with the principal and complementary subunits colored in pink and blue, respectively, and bound cardiolipin and PG shown as spheres (center). Zoomed in views of the bound cardiolipin and PG are shown on the left and right, respectively, with coordinating residues represented as sticks, with basic residues colored in blue, aromatic residues in yellow, polar residues in green and proline in tan.

## Data Availability

This study did not report any new data.

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
