# Peer review of "Recent Insight into Lipid Binding and Lipid Modulation of Pentameric Ligand-Gated Ion Channels"

_biomolecules, 2022, doi:10.3390/biom12060814_

Round 1
Reviewer 1 Report
This is a timely and detailed review on a topic that is receiving increasing attention. Lipids were mostly ignored in the early days of characterising ion channels but more recent data has shown that they can play a number of critical roles. This review concentrates on pLGICs and the authors have nicely described structures which have increasingly revealed the binding of specific lipids in specific locations, and have described functional studies that reveal the importance of lipids to receptor activation and modulation. My major criticism is that the recent work on lipids (especially cholesterol) at the NMJ nAChR by Unwin ( see below) has not been included, and these data could broaden and expand this section. The summary and conclusion section is also rather brief. It would enhance the review if the authors had provided their thoughts on the commonalities and differences of lipid effects in the range of pLGIC they have described, and also perhaps expanded this to compare with other ion channels, e.g K+ channels.
Unwin N (2020) Protein-lipid architecture of a cholinergic postsynaptic membrane. IUCrJ 7: 852–859.
Unwin N (2017) Segregation of lipids near acetylcholine-receptor channels imaged by cryo-EM. IUCrJ 4: 393–399.
Unwin N (2022) Protein-lipid interplay at the neuromuscular junction Microscopy, 71(S1), i66–i71
Minor points
Some english could be improved e.g line 99 ‘does not gate open’ would be better as ‘ does not open’ or ‘does not gate’
Line 668 ‘bound ‘ needs attention
The figures, which are great, could perhaps be more uniform in size ( they seem to get larger through the manuscript)
Author Response
We thank the reviewer for the evaluation of our manuscript. In what follows the reviewer’s comments are italicized our responses are in regular text.
My major criticism is that the recent work on lipids (especially cholesterol) at the NMJ nAChR by Unwin ( see below) has not been included, and these data could broaden and expand this section.
Unwin N (2020) Protein-lipid architecture of a cholinergic postsynaptic membrane. IUCrJ 7: 852–859.
Unwin N (2017) Segregation of lipids near acetylcholine-receptor channels imaged by cryo-EM. IUCrJ 4: 393–399.
Unwin N (2022) Protein-lipid interplay at the neuromuscular junction Microscopy, 71(S1), i66–i71
We agree that the recent work by Unwin adds valuable insight into cholesterol interactions – its absence was a serious omission from our review. We have included the following discussion about these studies (please see lines 218-235) and have added the appropriate references:
“It is notable that the cholesterol sites observed in the ?3?4, ?4?2 and Torpedo nAChR structures overlap with regions of low electron density in cryo-EM images recorded from native Torpedo post-synaptic membranes, with the low-density regions attributed to bound cholesterol [58-60]. The bound cholesterol is observed at both inner and outer leaflet transmembrane sites. Interestingly, the presence of cholesterol stabilizes a “splayed-apart” arrangement of the M1-M3-M4 a-helices in the outer leaflet of the bilayer, with this arrangement postulated to create space for the pore-lining M2 a-helices to move during gating [59,60]. Cholesterol-interacting regions become more extensive thus leading to the formation of microdomains in areas bridging adjacent receptors, particularly in the vicinity of the disulfide linkage between d-d dimers of neighboring nAChRs.”
The summary and conclusion section is also rather brief. It would enhance the review if the authors had provided their thoughts on the commonalities and differences of lipid effects in the range of pLGIC they have described, and also perhaps expanded this to compare with other ion channels, e.g K+ channels.
We appreciate the reviewer’s suggestion to expand this section. We have rewritten the entire section to clarify the overarching conclusions from the review. We did allude to differences in the levels of understanding of the mechanisms by which lipids modulate pLGICs versus other ion channels, such as inward rectifying potassium channels and mechanosensitive channels, but felt that further comparisons would require lengthy and detailed reviews of the mechanisms of lipid-protein interactions for these other channels. In attempting to rewrite this section incorporating this additional information, we realized that the inclusion of such details takes the Summary and Conclusions off in a tangential direction. We feel that the revised Summary and Conclusions section conveys effectively the key take home messages of the review, without getting lost in details of the underlying mechanisms.
“Our understanding of pLGIC-lipid interactions has exploded over the past decade. This understanding is increasingly shaped by new structures of pLGICs solved in different membrane environments. In fact, the plethora of new pLGIC structures solved over the past decade has begun to reveal both the complexities of lipid binding to pLGICs and the conformational transitions that underlie pLGIC function. Combining structural, functional and computational methods will eventually allow researchers to define precisely how lipids interact with pLGICs to preferentially stabilize one conformation over another to modulate pLGIC function. These multidisciplinary studies will also eventually lead to a detailed understanding of the role of pLGIC-lipid interactions in human biology.
Despite the enormous progress, there remain gaps in our knowledge. For most pLGICs, we still do not understand how lipids and bulk membrane properties influence channel gating and desensitization kinetics. We need better functional data on nanodisc-reconstituted pLGICs that will allow us to definitively assign solved structures to conformational states identified by electrophysiological methods. The vast toolbox of biochemical tools available for characterizing the function of the Torpedo nAChR should aid in this endeavor [41,59]. We also require a better understanding of how sample purification methods and different nanodisc preparations influence both pLGIC structure and the observed pLGIC-lipid interactions. Although our understanding of the mechanistic underpinnings of pLGIC-lipid interactions still lags behind that of other ion channels, such as inward rectifying potassium channels and mechanosensitive channels, where detailed models describing how signaling lipids and/or bulk membrane properties lead to channel activation have emerged [7], we are certainly at the dawn of a new age where will finally begin to understand the mechanistic underpinnings of pLGIC-lipid interactions.”
Some english could be improved e.g line 99 ‘does not gate open’ would be better as ‘ does not open’ or ‘does not gate’
We have corrected this error, the issue on line 99, which is now line 114, has been changed to “does not open”
Line 668 ‘bound ‘ needs attention
We have corrected this error, line 668 has been modified such that “bound” is no longer separated into two paragraphs
The figures, which are great, could perhaps be more uniform in size ( they seem to get larger through the manuscript)
We thank the reviewer for this suggestion. The variability in size of the figures is resulted from the formatting imposed by the journal. We agree that some of the figures, which were of lesser detail, were too large, so we changed the format of figures to arrange panels horizontally. This led to figures that fit better the format of the journal. Additionally, for figures 5 and 6 we have included only the transmembrane domain of the respective ion channel structures with zoomed in views arranged horizontally to better highlight the lipid sites and make the figures a more consistent size.
Reviewer 2 Report
This review article is from a leading scientist's group in the area of lipid binding and modulation of pentameric ligand-gated ion channels ''pLGICs. The subject matter is highly significant, both because of the critical roles of these pLGICs in human physiology and diseases, and because lipid binding/modulation has implications in pLGIC function and pharmacology. The text is well structured, the concepts are succinctly and clearly expressed, and the illustrations of pLGIC structures with lipids bound are of high quality. Readers of this review will learn what have been gained in the exploration of functional impacts and sites of binding of lipids in pLGICs, state-of-the-art methodologies utilized in these explorations and importantly, the knowledge gaps in this research area. For this review who does not do cryoEM experiments, the most striking information is that the sample preparation methods and the types of nanodiscs used in cryoEM can drastically influence the resulting structures.
Major issue:
There is a recurring theme in this review article: the inability to assign experimentally determined structural models to physiological functional states of pLGICs hampers the studies of how lipid binding modulates pLGIC function and the mechanism of such modulation. This limitation is astounding given the fact that most of the studies discussed in the review were done with state-of-the-art methodologies: cryoEM- and x-crystallographic high-resolution structures, molecular dynamics simulations, and functional assays.
Finally at the end, the authors gave us some hope: ... better functional data on nanodisc-reconstituted pLGICs that will allow us to definitively assign structures to physiological states (lines 724-726). Could the authors elaborate this statement: methodologies to obtain functional data on nanodisc-reconstituted pLGICs, references for such applications (even in other fields unrelated to pLGICs)?
Minor issues:
A few typos: line 333, 'of' should be 'or', pine 476 'a' should be 'an'.
Missing reference: lines 519-522, regarding PIP2 binding and functional roles.
Author Response
We thank the reviewer for the evaluation of our manuscript and for the positive evaluation of our manuscript. In what follows the reviewer’s comments are italicized our responses are in regular text.
Major issue:
There is a recurring theme in this review article: the inability to assign experimentally determined structural models to physiological functional states of pLGICs hampers the studies of how lipid binding modulates pLGIC function and the mechanism of such modulation. This limitation is astounding given the fact that most of the studies discussed in the review were done with state-of-the-art methodologies: cryoEM- and x-crystallographic high-resolution structures, molecular dynamics simulations, and functional assays.
Finally at the end, the authors gave us some hope: ... better functional data on nanodisc-reconstituted pLGICs that will allow us to definitively assign structures to physiological states (lines 724-726). Could the authors elaborate this statement: methodologies to obtain functional data on nanodisc-reconstituted pLGICs, references for such applications (even in other fields unrelated to pLGICs)?
We definitely appreciate the comments of the reviewer. The reviewer is absolutely correct in her/his assessment of our conclusions, but we have a slightly different take on this issue. Our sense is that there has been an explosion of new data in the past 10 years. We are now at a more difficult, but perhaps even more exciting phase where researchers can begin to pull together the diverse structural data into a unifying understanding. The ability to functionally characterize nano-disc remains, however, a major issue in the structural biology of ion channels. We have revised the Summary and Conclusions in response to the reviewer’s comments as follows:
“Our understanding of pLGIC-lipid interactions has exploded over the past decade. This understanding is increasingly shaped by new structures of pLGICs solved in different membrane environments. In fact, the plethora of new pLGIC structures solved over the past decade has begun to reveal both the complexities of lipid binding to pLGICs and the conformational transitions that underlie pLGIC function. Combining structural, functional and computational methods will eventually allow researchers to define precisely how lipids interact with pLGICs to preferentially stabilize one conformation over another to modulate pLGIC function. These multidisciplinary studies will also eventually lead to a detailed understanding of the role of pLGIC-lipid interactions in human biology.
Despite the enormous progress, there remain gaps in our knowledge. For most pLGICs, we still do not understand how lipids and bulk membrane properties influence channel gating and desensitization kinetics. We need better functional data on nanodisc-reconstituted pLGICs that will allow us to definitively assign solved structures to conformational states identified by electrophysiological methods. The vast toolbox of biochemical tools available for characterizing the function of the Torpedo nAChR should aid in this endeavor [41,59]. We also require a better understanding of how sample purification methods and different nanodisc preparations influence both pLGIC structure and the observed pLGIC-lipid interactions. Although our understanding of the mechanistic underpinnings of pLGIC-lipid interactions still lags behind that of other ion channels, such as inward rectifying potassium channels and mechanosensitive channels, where detailed models describing how signaling lipids and/or bulk membrane properties lead to channel activation have emerged [7], we are certainly at the dawn of a new age where will finally begin to understand the mechanistic underpinnings of pLGIC-lipid interactions.”
Minor issues:
A few typos: line 333, 'of' should be 'or', pine 476 'a' should be 'an'.
The type on line 333 has been corrected to read “or”. note that on line 531, we changed the “a” to an “an” so that it reads “an eukaryotic…”, but the editing software gave us a grammatical error. The software suggests that “a eukaryotic…” is the correct grammar
Missing reference: lines 519-522, regarding PIP2 binding and functional roles.
We have added an additional reference (reference 82) which suggested the role of PIP2 functional roles on GABAAR.